# Ultrasound and Photoacoustic Imaging of Breast Cancer: Clinical Systems, Challenges, and Future Outlook

**DOI:** 10.3390/jcm11051165

**Published:** 2022-02-22

**Authors:** Karl Kratkiewicz, Alexander Pattyn, Naser Alijabbari, Mohammad Mehrmohammadi

**Affiliations:** 1Department of Oncology, Wayne State University, Detroit, MI 48202, USA; fg9716@wayne.edu; 2Department of Biomedical Engineering, Wayne State University, Detroit, MI 48202, USA; apattyn@wayne.edu (A.P.); nalijabbari@wayne.edu (N.A.); 3Department of Electrical and Computer Engineering, Wayne State University, Detroit, MI 48202, USA; 4Barbara Ann Karmanos Cancer Institute, Detroit, MI 48202, USA

**Keywords:** breast cancer, breast imaging, screening, diagnostic imaging, ultrasound, photoacoustic imaging

## Abstract

Presently, breast cancer diagnostic methods are dominated by mammography. Although drawbacks of mammography are present including ionizing radiation and patient discomfort, not many alternatives are available. Ultrasound (US) is another method used in the diagnosis of breast cancer, commonly performed on women with dense breasts or in differentiating cysts from solid tumors. Handheld ultrasound (HHUS) and automated breast ultrasound (ABUS) are presently used to generate reflection images which do not contain quantitative information about the tissue. This limitation leads to a subjective interpretation from the sonographer. To rectify the subjective nature of ultrasound, ultrasound tomography (UST) systems have been developed to acquire both reflection and transmission UST (TUST) images. This allows for quantitative assessment of tissue sound speed (SS) and acoustic attenuation which can be used to evaluate the stiffness of the lesions. Another imaging modality being used to detect breast cancer is photoacoustic tomography (PAT). Utilizing much of the same hardware as ultrasound tomography, PAT receives acoustic waves generated from tissue chromophores that are optically excited by a high energy pulsed laser. This allows the user to ideally produce chromophore concentration maps or extract other tissue parameters through spectroscopic PAT. Here, several systems in the area of TUST and PAT are discussed along with their advantages and disadvantages in breast cancer diagnosis. This overview of available systems can provide a landscape of possible intersections and future refinements in cancer diagnosis.

## 1. Introduction

As of 2020, breast cancer has surpassed lung cancer as the most diagnosed cancer worldwide, reaching 2.3 million new cases [1]. The number of patients who passed away due to breast cancer globally was 685,000 in 2020 [2]. This is also a prevalent issue in the U.S. as The American Cancer Society reported that about 1 in 8 women will be diagnosed with invasive breast cancer in their lifetime and a total of 44,130 women will die from breast cancer. Early detection of breast cancer can play a vital role in reducing the mortality rate of breast cancer, seeing as since its peak in 1989 the female breast cancer death rate has fallen by 41% in the U.S. as of 2018.Therefore, reliable screening of asymptomatic women is critical for saving lives. Presently, mammography and US are routinely used for screening. Both tools provide morphological information about the underlying tissue.

In mammography one or more two-dimensional X-ray projections of the compressed breast are created [3]. Limitations of mammography include the high false positive rate of breast cancer in radiographically dense breasts [4,5,6]. The breast is composed of mainly fat and fibroglandular tissue. Fat is radiologically transparent, whereas the fibroglandular tissue and masses can both appear dense on a mammogram [7]. Dense breasts composed of greater amounts of fibroglandular tissue and therefore have poor contrast between the fibroglandular tissue and a potential tumor [7]. Approximately half of all women who are ≥40 years old and receive mammograms are discovered to have dense breast [8]. Some state laws require that women with dense breasts be made aware of additional screening available [9]. Close to 12% of women screened with digital mammography currently need follow-up or biopsy and 95% of these women will be found to not have cancer. There is room to improve the accuracy of breast cancer screening methods.

Ultrasound (US) uses the reflected acoustic pressure from different tissue interfaces originated from piezoelectric elements in the transducer linear array to create an image of the breast [10]. US can be used for further examination following mammographic evidence of dense breast; however, US has low soft tissue contrast [11], low specificity when compared to mammography [12], and is operator dependent (in the case of handheld US) [13]. Although used less frequently for breast cancer diagnosis, another diagnostic imaging system is magnetic resonance imaging (MRI), which can provide a three-dimensional volumetric image of the breast without ionizing radiation. Conversely, it cannot be part of routine screening since it requires a contrast agent, is more time consuming, and expensive when compared to mammography and US [14]. These drawbacks of both mammography and US prompt the need for improved breast cancer diagnostic methods, especially in the case of radiographically dense breast.

As described, traditional US utilizes a linear array of piezoelectric elements which generate acoustic pressure waves that propagate through the tissue. As these waves interact with changes in media, reflected signals are sent back toward the ultrasound probe where the received echoes can be back projected to form the reflection image [10]. Transmission ultrasound tomography (TUST) goes one step further to not only utilize information from reflected signals, but also transmission signals by incorporating piezoelectric elements on opposing sides of the media of interest. Examples of these types of geometries include ring arrays [15], opposing arrays [16], or hemispherical arrays [17]. The inclusion of transmission signals allows the user to extract more quantitative parameters from the tissue to be included in diagnostic assessment. For instance, tissue sound speed (SS) is mapped based on the time-of-flight of the waves as they cross from one side of the media to the other [18]. Additionally, tissue attenuation can be mapped based on the changes in signal amplitude [19]. The ability to map these tissue parameters makes TUST systems attractive as now cancerous masses can be localized based on a combination of tissue SS, attenuation, and structural information of the reflection image. Reflection images will provide tissue edge delineation, allowing clinicians to visualize potential regions of interest through structural anomalies. Tissue SS and attenuation provide two more metrics for differential diagnosis of suspicious tissue, whether healthy (normal SS and attenuation), benign (heightened SS, low attenuation), or malignant (heightened SS and attenuation) [19]. In addition, the combination of these imaging modes will provide clinicians an estimate of breast tissue density; an important risk factor in anticipating future development of breast cancer [20].

A natural adjunct for UST is photoacoustic (PA) tomography (PAT), which is—in part—based on the same fundamental principles as UST. PAT can provide functional and molecular information by capitalizing on the photoacoustic effect. This phenomenon uses light to excite both endogenous and exogenous chromophores, which absorb that optical energy and undergo the process of thermal elastic expansion, where a portion of that energy is converted into acoustic pressure waves. It is through these acoustic waves that a natural synergy originates since these waves can be detected with a typical US transducer array [21]. The generation of these acoustic waves can be described by Equation (1):p_0_ = ΓFµ_a_= ΓA_e_(1)
where p_0_ is the photoacoustic pressure field after excitation, Γ is the Grueneisen parameter, F is the light fluence, μ_a_ is the optical absorption, and A_e_ is the optical energy density [22]. From Equation (1), we can see that the acoustic pressure field, i.e., the reconstructed PA image is proportional to the optical absorption coefficient, which is the source of contrast in PA images and from where that functional and molecular information are derived. Due to the wavelength dependent nature of PA imaging, it can be used in spectroscopic imaging to generate quantitative information such as oxygen saturation (sO_2_) [23,24,25] which is an important indicator in cancer progression. Therefore, PA imaging is a potentially powerful adjunct to US imaging, offering more indicators that can be used to form a potentially more informed diagnosis in the progression of cancer and its treatment. 

In this review, several TUST and PAT systems currently performing clinical trials or being researched preclinically are summarized. The TUST systems presented are varied in design. They include systems utilizing a ring transducer, such that the breast is vertically scanned to acquire three-dimensional images, to rotated transducer designs, and, finally, a multi-element, hemispherical transducer. The PAT systems are also varied in their construction: with breast compression based illuminated systems, to rotating laser systems, and finally, diffuse source illumination systems. Alongside current PAT systems, the role of exogenous PA contrast agents as they contribute to breast cancer detection is examined. Finally, the potential future of the combination of these modalities into a single system toward full breast ultrasound/photoacoustic tomography (USPAT) imaging is discussed. We hope that presenting the advantages and disadvantages of these systems along with our perspective on the future of USPAT can provide the reader with a landscape of systems which can lead to intersections and future refinements for cancer diagnosis.

## 2. Transmission Ultrasound Tomography Systems

Here, a few TUST systems currently performing in vivo clinical tests are discussed. They include Delphinus Medical Technologies’ SoftVue system, QT Ultrasound’s QT Scanner 2000, Mastoscopia’s MUT Mark II system, and Karlsruhe Institute of Technology’s KIT 3D USCT system. These systems operate in a similar fashion toward full breast cancer detection; however, they do this in different ways.

The Delphinus Medical Technologies’ SoftVue system uses a ring transducer with an inner diameter of 22 cm. The ring is comprised of 2048 elements and has a center frequency of 2.5 MHz. Central frequency is a critical component of TUST as a higher central frequency improves reflection mode resolution. Bandwidth is critical in TUST as a wider bandwidth will result in a shorter (time-domain) transmit pulse that is more easily recognized for travel time estimation in SS imaging. This improves the SS image accuracy. During data acquisition, a single element transmits a planar wave while all the other elements record. Each element successively fires in the same fashion for a given elevational slice. The ring is then translated vertically so that 20–80 cross-sectional slices of data are acquired per breast in order to create a volumetric image. SoftVue can generate reflection, SS and attenuation images and provide volume averaged speed of sound (VASS) values for breast density stratification. The SoftVue system is shown in Figure 1. Figure 1a shows a schematic of the vertically translatable ring encompassing the breast. Figure 1b graphically shows the single element transmission and all other element reception method of data acquisition that the system follows. Figure 1c previews an example output coronal SS image of a female breast from the SoftVue system where darker pixels represent lower SS and brighter pixels represent higher SS. The sub-millimeter spatial resolution captures the irregular shape of the cancer at 8 o’clock, as well as parenchymal detail (resolution: 0.7 mm). Finally, Figure 1d shows the entire system [20,26].

The QT Ultrasound scanner 2000 system is described next which, rather than a ring array, uses opposing arrays for transmission imaging and three other arrays for reflection imaging to create what they call a U-channel. The opposing arrays, therefore, are used to generate the SS and attenuation maps, whereas the separate arrays generate the reflection image. In this system, the 256-element array transmitter emits a wideband chirp between 0.3 and 1.5 MHz (0.9 MHz central) which travels to a 2048 element 2D receiver array (8 rows by 256 columns or 20 mm × 128 mm). The height of the 2D receiver array allows for out of plane data collection accounting for 3D refractive effects along with a 3D back projection (correcting for small angles) and 3D forward simulation algorithm. At the same time, the three reflection arrays have a central frequency of 3.6 MHz and varying focal depths of 25, 45, and 75 mm to generate the reflection images. A full cross section requires a 360-degree mechanical rotation of the scanning array that is then followed by vertical mechanical translation to produce volumetric images. The QT Ultrasound scanner 2000 can provide basic breast density based on dense and non-dense tissue groupings. Figure 2 shows the QT Ultrasound scanner 2000. Figure 2a shows the entire scanner system in which the patient lies prone. Figure 2b shows a 2D schematic of the ultrasound hardware of the system where the transmission transmitter array, transmission receiver array, and reflection arrays are identified. Figure 2c shows an example output SS image from the QT Ultrasound scanner 2000, where darker pixels represent lower SS and brighter pixels represent higher SS [27]. Figures adopted from [27].

The Mastoscopia MUT Mark II system uses an opposing transmitter and receiver array geometry where the arrays are 24 cm apart and each are 2 MHz central frequency. The system performs a 180-degree rotation constantly recording transmission and reflection signals. The assembly is then translated vertically by 4 mm between breast slices until the entire breast volume is scanned. Unlike the previous two systems, the MUT Mark II system does not output reflection mode, SS, and attenuation images. Instead, the system extracts the acoustic attributes: refractivity index, attenuation, and dispersion to be plugged into a specially designed algorithm that outputs a probability of malignancy map. This is a color composite image with blue being the least malignant to red being the most malignant. A stack of images can then be reviewed by trained personnel to check for possible anomalies or cancer. The MUT Mark II system’s malignancy results are reported to match well with biopsy and histopathology data; however, no attempts are made to produce a quantitative density number like the previous two systems. Figure 3 shows the Mastoscopia MUT Mark II system. Figure 3a shows the full system in which the patient lies prone. Figure 3b shows a schematic of the ultrasound geometry in which two opposing linear arrays circle the patient’s breast. Figure 3c shows an example output of the probability of malignancy map from the system [16]. Figures adopted from [16,28].

The Karlsruhe Institute of Technology’s KIT 3D USCT system presented next is the only system that can be said to truly be a 3D tomographic system. It is comprised of a hemispherical array geometry using 628 transmitters and 1413 dedicated receivers arranged around the breast. Each transducer used has a 2.5 MHz center frequency. Similar to the Delphinus system, each transmitter independently fires while all receivers record. The number of transducer locations can be further increased by rotating the hemisphere system to result in −3.5 million A-Scans per single breast. Like Delphinus and QT systems, a single image acquisition allows reconstruction of reflectivity, SS and attenuation maps. Figure 4 demonstrates the KIT 3D USCT system. Figure 4a shows the full system where the patient lies prone along with a real image of the hemispherical transducer array. Figure 4b shows an example fusion reflectivity, SS, and attenuation image [17]. Figures adopted from [17,29].

The TUST systems described here are not meant to be all encompassing, but representative of common methodologies used by groups pushing the field of ultrasound to improve methods for breast cancer diagnosis. By providing quantitative information, such as SS, attenuation, and density scores, these systems provide additional information rather than just traditional reflectivity that can aid in patient diagnosis. Table 1 below shows a list of different features of the discussed systems. Some advantages and limitations of the reviewed systems to consider include bandwidth, geometry, and computation time. The bandwidth of the transducer elements relates to how short (time-domain) the transmit pulse will be. A shorter pulse from a larger bandwidth transducer will result in a more accurate time-of-flight estimate when analyzing the receive data for SS inversion. Further, geometry will result in certain advantages such as avoiding additional complexity of rotation in Delphinus’ case compared to QT Ultrasound and Mastoscopia. Geometry also has advantages where the Karlsruhe system has an intrinsic 3D nature, but the computation time will be greater due to the heavy burden of inverting a full 3D volume. It is also important to consider the differences in size of detectable lesion between systems This will be governed by the resolutions of the systems, where higher resolution will more accurately detect smaller lesions.

## 3. Photacoustic Imaging Systems

Below various PA systems are discussed, each of which approaches the problem of functional breast cancer imaging differently. Among this non-exhaustive list are the Kyoto-Cannon photoacoustic mammography (PAM)-02, Twente PAM, OptoSonics PAM, Laser Optoacoustic Ultrasonic Imaging System Assembly (LOUISA-3D), the single-breath-hold PA computed tomography (SBH-PACT) system, and the Seno Medical Imagio system. They vary in overall design to adjust how the breast is illuminated to generate photoacoustic signals. Each, however, pushes the field of photoacoustic breast cancer imaging forward.

The Kyoto-Canon PAM-02 system is a US-PA combined imaging system based on the general design of a traditional mammography system. Figure 5a shows a schematic of the system, where two polymer plates are used to slightly compress the breast at which point US and PA image acquisition occurs. Due to the projection images generated by PAM-02, the patient’s breast must be imaged at multiple positions—which can be seen in Figure 5b. The polymer used for compression are made of two materials, (1) on the transducer side polymethylpentene (PMP) is used due to its acoustic attenuation and wave velocity values –4.4 dB/cm at 4 MHz and 2.22 mm/µs, and is 4 mm thick, while (2) is on the side of the laser beam that illuminates the breast tissue and is made of polymethyl methacrylate (PMMA) which was selected for its high light transmittance at a thickness of 23 mm. In order to couple the breast to the PMP a nanocomposite was used due to the orientation of the US-PA detector being vertical, if a more traditional ultrasound gel was used it would fall due to gravity. This nanocomposite can be compressed by the breast to form a good couple. In order to generate the photoacoustic signals a Titanium-Sapphire (Ti:Sa) laser is used and is optically pumped using a Q-switched neodymium: yttrium-aluminum-garnet (Nd:YAG) laser which is tunable between 700 and 900 nm. In order to produce hemoglobin saturation maps two wavelengths are used, these are 756 and 797 nm. The PAM-02 system also has three illumination modes: (1) backwards—from the PMP side, (2) forward—from PMMA side, and (3) both. The beam from the backwards direction is oblong with a size of −3 cm × 1 cm and follows two paths next to the transducer, while the forward beam is −3 cm × 4 cm. The transducer used to detect the PA signals has the following properties: 600 elements arranged in a matrix that is 20 × 30, center frequency of 2 MHz, and a fractional bandwidth of 130%. In PAT imaging, the PA source emits a broadband pulse which can be received by most transducers. However, using different frequency and bandwidth receivers will modify the frequency range of the source that is received, which will affect the resolution of imaging. The scanning is conducted in the x-y plane as defined by Figure 5a and after each pulse of the laser the transducer module moves in the horizontal direction, such that the same area of the breast is scanned 20 times; this is due to the PA detection having 20 elements in the horizontal direction. This allows for the potential of 20 averages per location on the surface of the breast, which can reduce the amount of noise within the PA images [30]. Figures adopted from [30].

The Twente PAM 2 system is another PA imaging system that uses 12 arch-shaped arrays for PA detection along with nine fiber bundles (see Figure 6). The PA detector is composed of a hemispherical array that contains 384 elements which are arranged in 12 arcs that follow the curvature of the breast (Figure 6a). These elements have a center frequency of 1 MHz and a fractional bandwidth of 100%. The illumination system is composed of a dual-laser system composed of a Q-switched Nd:YAG and Alexandrite laser whose pulse durations are 5 ns and 60 ns, respectively, and has a pulse repetition rate of 10 Hz in single wavelength mode and 20 Hz in double wavelength mode. The wavelengths used for imaging are 1064 and 755 nm, which Twente chose due to the predicted maximum light penetration for breast imaging (1064 nm) and increased hemoglobin absorption (755 nm). To image the patient’s breast, the transducer and fiber bundles are rotated around the breast, which allows for more views and increased sampling. As can be seen in Figure 6a the illumination occurs with nine fiber bundles which are interleaved with the 12 arc arrays. Figure 6b shows the full Twente PAM system. Figures adopted from [31].

Next, the PAM imaging system by OptoSonics is discussed. However, unlike the others above, the Optosonics systems utilizes a hemispherical design (see Figure 7). The hemisphere is controlled by a 2D stage that is moved by two synchronous motors. The hemisphere itself is made from acrylonitrile butadiene styrene (ABS) plastic with a radius of 127 mm and holds 512 elements that have a 2 MHz center frequency and a bandwidth of 70%. For PA imaging a 7 mm diameter pulsed Alexandrite laser beam is directed upward from the bottom of the hemisphere through a 12 mm diverging lens and then onto the patient’s breast. The wavelengths that are typically used are between 750–800 nm. The hemisphere is continuously scanning in a spiral pattern, such that all the PA data is acquired at equidistant points. The maximum imaging volume measured in the system is 1335 mL, which is a function of the size of the hemisphere as well as the imaging depth of the system [32]. Figures adopted from [32].

Another unique design when compared to the more traditional compression mammography imaging systems described in this review, is the LOUISA-3D—see Figure 8. The LOUISA-3D consists of a hemispherical mold for the patient’s breast, an arc shaped transducer array, and an arc shaped fiber bundle. These arc shaped transducer and fiber arrays rotate around the breast in order to provide volumetric imaging. The transducer module consists of 96 elements with a frequency range between 50 kHz–6 MHz. The laser used for imaging is a dual wavelength pulsed Alexandrite laser, which fires at 757 and 797 nm. Each wavelength is separated by approximately 50–100 ms delay, which allows for accurate co-registration PA images for the generation of oxygen saturation maps. This beam is fed through an arc shaped fiber bundle to provide homogeneous illumination. After the PA system has made one full scan, there are 34,560 data points—“virtual transducers”—for image reconstruction. Additionally, the LOUISA-3D has an US subsystem that consists of an arc array with 192 elements, which operate at a 7 MHz center frequency with a 100% bandwidth. This subsystem allows for 2D slices of the breast and can be overlayed with the PA images [33]. Figures adopted from [33,34].

Next, the SBH-PACT is covered (see Figure 9). This system uses a modified version of the traditional mammography design, such that the breast is placed between a ring array transducer and compressed from the nipple to the chest wall using an agar pillow. The coupling medium is preheated water (35 °C). A linear stage is present to allow the ring array to scan the full volume of the breast. The ring array transducer is composed of 512 elements with a diameter of 220 mm, center frequency of 2.25 MHz, and a bandwidth of 95%. The illumination system is comprised of a Quanta-Ray 1064 nm laser beam with a pulse repetition rate of 10 Hz and a pulse width of 8–12 ns. In order to acquire oxygen saturation maps, an additional wavelength of 750 nm is used [35]. The reported penetration depth of the system is 4 cm, which would allow it to fully image breast that are 8 cm in diameter. Figures adopted from [35].

The last PA imaging system covered in this review is Imagio by Seno Medical, which has received Food and Drug Administration (FDA) premarket approval (PMA). Unlike the PA systems described above, Imagio is comprised of a handheld ultrasound and photoacoustic probe with an integrated optical fiber bundle (see Figure 10). Imagio uses a dual wavelength laser system, where the 1064 nm wavelength is produced from a Nd:YAG laser with a 15 ns pulse width with a pulse energy 150 mJ, while the 757 nm is from a Alexandrite laser with a 50 ns pulse width with a pulse energy of 140 mJ. In order to co-register the two wavelengths and to minimize motion artifacts there is a 5 ms delay between switching wavelengths. These laser beams are carried via an optical fiber bundle, that is made up of 200 µm diameter fibers that are then spread uniformly across two 40 mm × 6 mm optical windows that also contain as a part of the optical assembly custom lenses and optical diffusers. This combined US/PA probe was optimized for tumors with a diameter of ~10 mm, however, objects that range from 3–20 mm can be visualized. Additionally, objects like large blood vessels could be visualized up to depths of 40 mm. While the Imagio lacks the advantages of UST, such as SS and acoustic attenuation measurement, it can reconstruct US/PA images in real-time, which can be advantageous for real time diagnosis [36,37,38,39]. Figures adopted from [37,38].

As with the TUST section, the PA systems described here are not meant to be all encompassing, but representative of common methodologies used by groups pushing the field of photoacoustics to improve methods for breast cancer diagnosis. All of these systems aim to provide quantitative information relating to hemoglobin, such as oxygenation saturation, and vessel density which provide additional clinically valuable information for screening, differential diagnosis, and determining the stage of cancer [37]. Some of the aforementioned systems use particular wavelengths depending on their targeted contrast agents and laser type, but the Twente PAM 2, SBH-PACT and Imagio make use of 1064 nm. This is of particular importance due to it being a principle wavelength of Nd:YAG lasers which allows it to be emitted at high energies and allows for increased penetration depth [40]. Imaging platforms that use Nd:YAG lasers can maintain the relevant low-cost nature of US and PA, while also having access to a high energy wavelength and increased penetration depth which is important in applications like breast imaging. Table 2 below shows a list of different features of the discussed systems. Similar to TUST, the size of lesion detected will depend on resolution of the system. PAT resolution is governed in the same way as ultrasound where a higher frequency will have a better resolution. Geometry will also affect receive data coherence which may change resolution.

## 4. Exogenous Contrast Agents for Photoacoustic Molecular Imaging

The described PA systems use oxygenated and deoxygenated hemoglobin as the primary endogenous contrast agents during imaging. While this is important for tracking the transformation of vasculature during disease progression, it does potentially carry with it a higher contrast to noise ratio than exogenous contrast agents. Exogenous contrast agents can be conjugated and targeted to illuminate specific cancer sites or other pathologies, and thereby potentially increasing the sensitivity and specificity of PA imaging. This includes the ability to visualize the presence of masses smaller than the resolution limit of the hardware due to spreading of the received PA signals to the hardware resolution. While some of the PA systems described above only report using a limited range of wavelengths there is no hard limitation on the use of exogenous contrast agents for any of the described systems. As such, any of the PA imaging systems reviewed can use exogenous contrast agents [41,42,43,44,45]. While there may be some limitations regarding the range of wavelengths that each group uses, this can be solved by utilizing a tunable laser—at the expense of energy output and therefore penetration depth. As most illumination/light delivery systems—gold/silver coated mirrors or fiber optics—can support a wide range of wavelengths that are applicable for clinical imaging, i.e., the visible and near-infrared range. Therefore, all have the potential to allow for exogenous contrast agent enhanced PA imaging.

An example of an exogenous contrast agent is indocyanine green (ICG) conjugates, which when bound to an antibody can undergo dynamic absorption spectrum shifts after cellular endocytosis and degradation [46]. This, in turn, can be used to differentiate normal breast tissue from cancerous tissue by targeting receptors expressed on tumor cells. A second FDA-approved exogenous contrast agent is methylene blue, which like ICG allows for accumulation within tumors due to their small size and ability to bind to both endothelial and epithelial cell surface receptors [47]. Methylene blue has been used for imaging of sentinel lymph nodes (SLN), which are the first nodes in the lymphatic system that drains a tumor site. Typically, a negative SLN biopsy result suggests that cancer has not spread to nearby lymph nodes or other organs, and PA can determine if the lymph node is an SLN by imaging accumulated blue dye.

An alternative to these traditional contrast enhancing dyes are nanoparticles. Nanoparticles can be made from a variety of materials and manufactured into various shapes, which makes them highly tunable depending on their purpose [41]. A common choice is gold nanospheres due to the simplicity in manufacturing, and they have a peak absorption around 520 nm, which is useful as it lies within a local minimum in the optical window of both blood and skin [41,48,49]. Single-walled carbon nanotubes (SWNTs) also fall into this category of exogenous contrast agents [50]. Which have a wide absorption spectrum, outside the visible spectrum up to 3 GHz. Typically, SWNTs are used in conjunction with other contrast enhancing agents, such as gold nanorods, various dyes, or targeting agents like arginylglycylaspartic acid (RGD) peptides, which enhance the resulting PA signal than if they were used alone. The next advancement in using nanoparticles for contrast enhancement looks at being able to have more direct control of the release of these contrast agents through the use of thermo-responsive nanoparticles [51]. These thermo-responsive contrast agents allow for remote influence of PA signals via photoirradiation to enhance the signal-to-background noise of various targets, such as tumors. This allows for enhanced signal generation within a region-of-interest while ignoring the background. 

One advantage of PA imaging is its ability to use endogenous contrast agents to derive quantitative information on disease/pathology progression. However, when the background noise levels are too high, or when it is difficult to distinguish the region-of-interest from the surrounding tissue, alternative methods have been developed to enhance contrast. These contrast-enhancing exogenous agents can target specific pathologies more efficiently and increase the sensitivity and specificity of photoacoustic imaging. This makes PA an extremely adaptive imaging modality and powerful adjunct to UST, such that a combined system will be able to provide physicians invaluable insights into disease progression and screening.

## 5. Discussion and Conclusions

In this review, current technology being investigated both preclinically and in current clinical trials for both TUST and PAT were examined. TUST provides improved diagnosis of breast cancer through a multimodal approach of mapping tissue reflectivity, SS, and attenuation, or in the case of Mastoscopia, probability of malignancy. Compared to the structural information gleaned from TUST, PAT provides functional information through monitoring tissue endogenous contrast agent total concentration and the potential for oxygen saturation measurement. We also discussed the utility of exogenous contrast agents being used to localize malignancy through targeted molecular methods. We believe the future of breast cancer diagnosis is in the intersection of these two modalities in the form of an ultrasound/photoacoustic tomography (USPAT) system. As we have mentioned, these modalities share most hardware for signal detection which would result in inherently co-registered images combining structural and functional information in a bedside imaging modality. One limitation of PAT systems is in deep tissue differences in fluence between wavelengths. This can introduce error in accurately detecting tissue oxygenation. In a USPAT system, the SS and attenuation images can naturally be used in an optical forward model for fluence modeling that could directly compensate the photoacoustic images.

We have recently reported a combined US and PA tomography system based on a ring geometry [52]. In our developed breast USPAT (BUSPAT) imaging system, a 200 mm diameter, 256-element ring US transducer (Sound Technology (STI), State College, PA) was used with a center frequency of 1.5 MHz and bandwidth of 60% at a sampling rate of 8.33 MHz to record all ultrasound and photoacoustic data. Photoacoustic data was generated using a tunable 10 ns pulsed laser (Phocus Core, Optotek, Carlsbad, CA, USA) along with two custom ring mirrors from Syntec Optics (Rochester, NY, US) with a 10 mm diameter axicon mirror (68–791, Edmund Optics, Barrington, NJ, USA) to provide full-ring illumination on a phantom. Reflection and SS images were obtained through the same means as described with the Delphinus SoftVue system and were used in PAT image compensation. We incorporated SS compensation in the actual back projection of the PAT receive waveforms by back projecting onto the family of iso-time-of-flight contours instead of concentric circles found via a straight ray approximation of the original SS image obtained. Fluence compensation was incorporated via the SS image as a priori structural information so that a seeded region growing method could segment and apply optical properties to a given phantom. A Monte Carlo light simulation through MCXCL was then run modeling the full-ring illumination scheme we implemented. The inverse of the output fluence map was then multiplied to the original SS compensated PAT image. Figure 11 shows the combined USPAT system prototype. Figure 11a shows the schematic of the system. Figure 11b shows the heterogeneous phantom design. Figure 11c shows the original SS image, PAT image after just SS compensation, and PAT image after both SS and fluence compensation. Figures adopted from [52].

We demonstrated that the combination of these two imaging modalities has improved the feasibility of deep tissue functional imaging with PAT. Still, this system has limitations. Specifically, we are using a ring array with a limited number of transducer elements, which is hindering the quality of our output SS images. As the initial SS image is critical in the compensation methods for improving the PAT images, this is a large drawback. This can be improved with more complex hardware. However, there is still intrinsic problems in the SS reconstruction algorithm. The SS reconstruction infers a 2D wave propagation which is flawed due to the 3D reality of the ring array geometry. 3D artifacts in the SS algorithm must also be resolved by implementing a wholly 3D reconstruction which appropriately models the beamforming being performed during ultrasound transmission.

As discussed, there is a need to improve breast cancer diagnostic methods due to difficulty in confirming malignancy in radiographically dense breast. Traditional US lacks soft tissue contrast and is user dependent; however, there is much study on advancement of TUST systems. These will increase the number of mapped tissue properties to improve the ability of localizing malignancy. Further, PAT can operate as an adjunct modality to TUST using the same hardware but providing functional information like endogenous chromophore concentration or oxygen saturation. Exogenous contrast agents can also be used as a molecular imaging method of localizing malignancy. Yet there are limitations in PAT’s ability to accurately image oxygen saturation due to differences in optical fluence in deep tissues between wavelengths. Therefore, the future of breast cancer detection is a combined USPAT imaging modality with the direct next steps being to increase the number of elements in the ring array and to sophisticate the image reconstruction algorithm to appropriately model the 3D effects of transmission wave propagation.

## Figures and Tables

**Figure 1 jcm-11-01165-f001:**
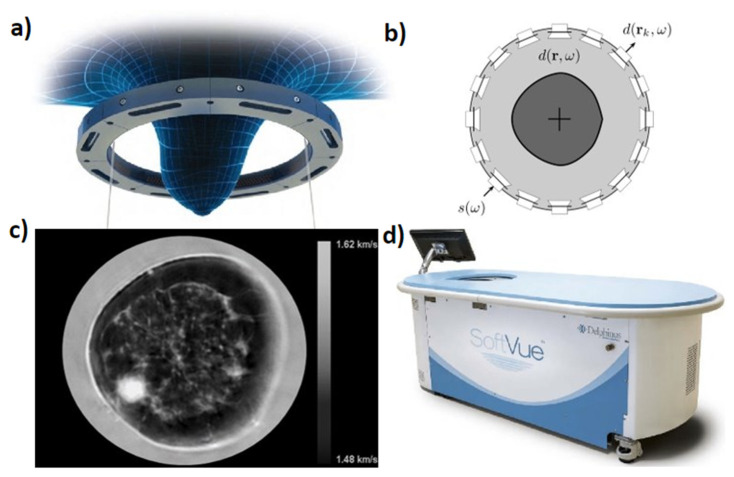
Delphinus Medical Technologies’ SoftVue system (**a**) Ring transducer used for data acquisition, (**b**) transducer ring configuration, (**c**) example of a sound speed (SS) cross sectional slice with darker pixels representing lower SS and brighter pixels representing higher SS, (**d**) the clinical system.

**Figure 2 jcm-11-01165-f002:**
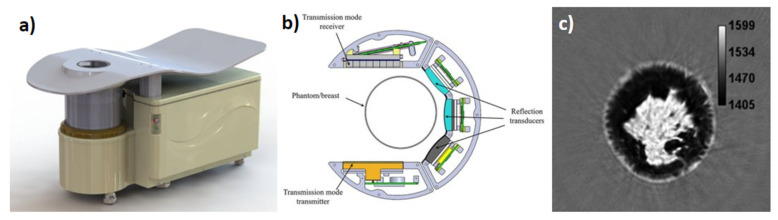
QT Ultrasound scanner 2000 (**a**) the full clinical system, (**b**) top view of reflection and transmission array platform, (**c**) example of an SS cross sectional slice with darker areas representing lower SS and brighter pixels.

**Figure 3 jcm-11-01165-f003:**
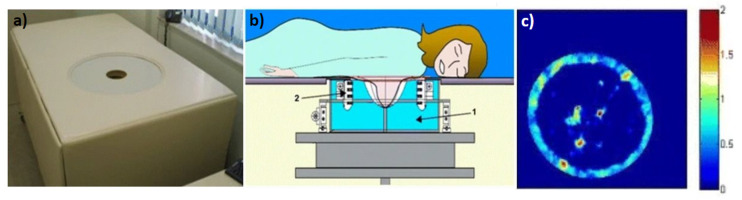
Mastoscopia’s MUT Mark II system (**a**) the full clinical system, (**b**) rotating transmitter and system assembly used for imaging showing (1) scanning chamber and (2) transmitter/receiver array, (**c**) color composite image generated by using ‘refractivity’, attenuation and dispersion values with adipose tissue and normal breast parenchyma producing lower values less than 1, while tumors would produce values closer to 2.

**Figure 4 jcm-11-01165-f004:**
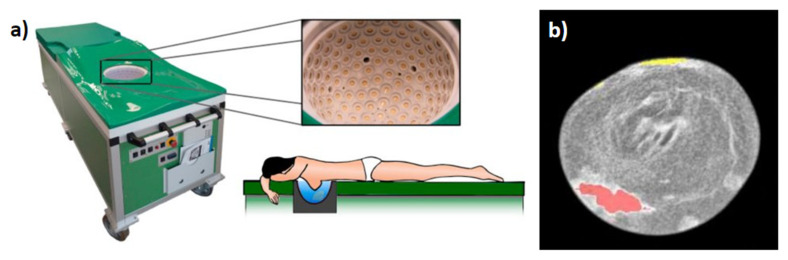
Design by Karlsruhe Institute for Technology: (**a**) the clinical system showing patient laying prone along with close-up of ellipsoid tank with transducer array elements each containing four emitters and nine receivers, (**b**) example fusion reflectivity, SS, and attenuation image. Red and yellow overlay denoting suspected malignant and benign tissue, respectively.

**Figure 5 jcm-11-01165-f005:**
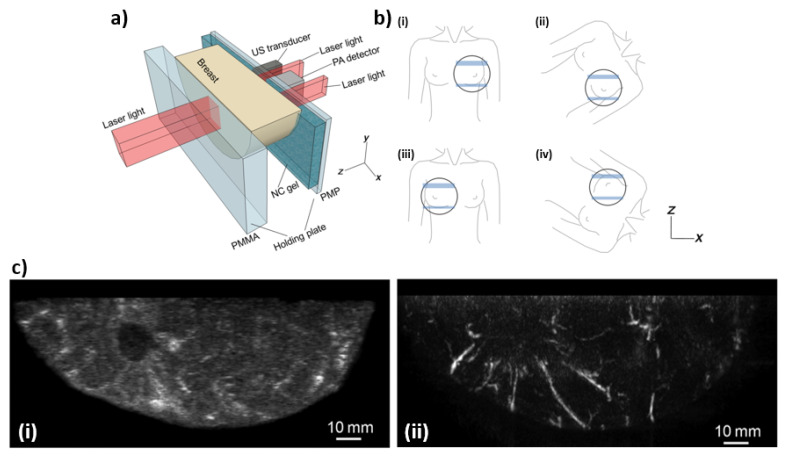
(**a**) Schematic illustration of the breast-holding structure. (**b**) Schematic illustration of breast-holding direction: (**i**) cranio-caudal (CC) of left breast, (**ii**) medio-lateral-oblique (MLO) of left breast, (**iii**) CC of right breast, and (**iv**) MLO of right breast. (**c**) Lesion images of a breast in (**i**) ultrasound and (**ii**) photoacoustic modes.

**Figure 6 jcm-11-01165-f006:**
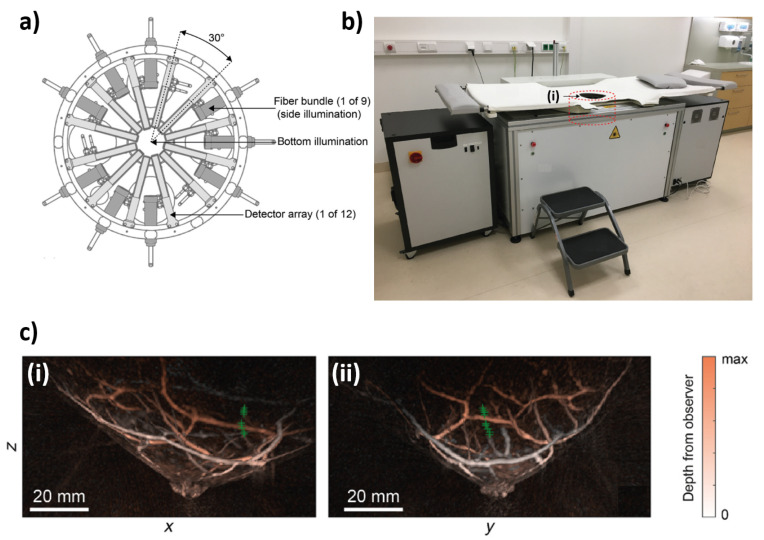
(**a**) Schematic of the Twente PAM 2 imaging tank. (**b**) Photograph of the Twente photoacoustic mammography (PAM) 2 system—(**i**) is the imaging tank. (**c**) Local maximum intensity projections of a breast in the (**i**) sagittal and (**ii**) transverse plane.

**Figure 7 jcm-11-01165-f007:**
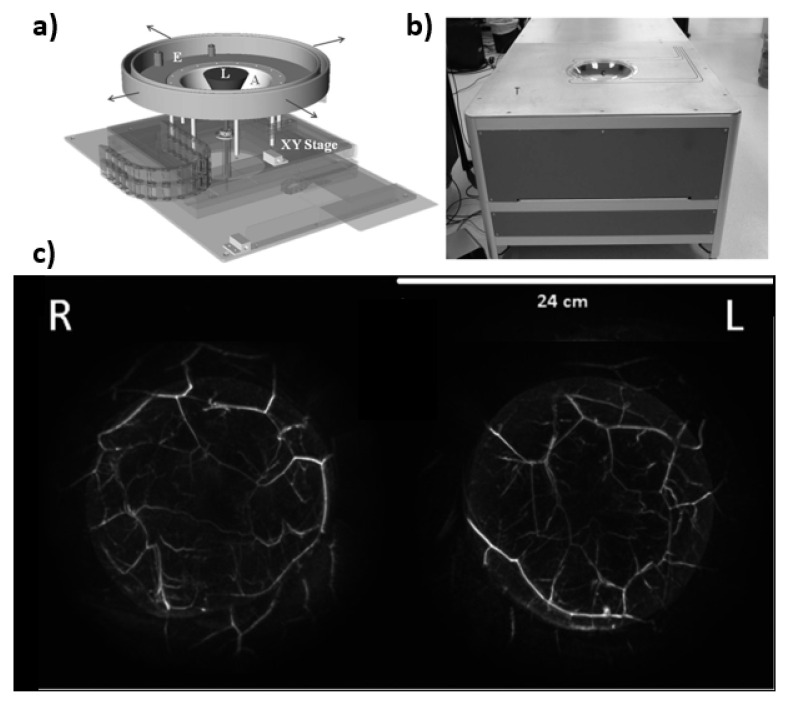
(**a**) Drawing showing the hemispherical array (A) mounted on a two-axis translational stage (XY). The hemispherical array and an extension (E) are filled with degassed water. Laser light is fed from the bottom of the array via an articulating arm (not shown) through a negative lens that diverges the laser light (L) to a diameter of −60 mm at the breast surface. (**b**) Photograph of PAM scanner showing the exam table (T) and the breast positioning cup (C), below which is located the hemispherical detector array. (**c**) Maximum amplitude projection images of a breast for both right (**R**) and left (**L**) sides.

**Figure 8 jcm-11-01165-f008:**
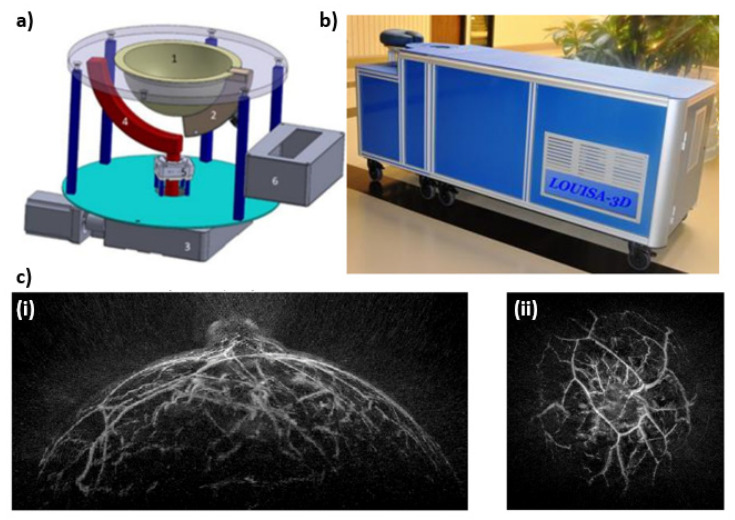
(**a**) Imaging module capable of scanning large breast consisting of (1) Imaging bowl that contains acoustic coupling medium and breast, (2) Optoacoustic transducer array, (3) Large motor to rotate the entire imaging module around the breast, (4) Arc-shaped optical fiber to illuminate the breast with homogeneous beam of light, (5) Small motor to rotate the arc-shaped fiberoptic paddle around the imaging bowl, and (6) preamplifier boards directly connected to the probe. (**b**) Photograph of the Laser Optoacoustic Ultrasonic Imaging System Assembly (LOUISA-3D) system designed as an examination bed. (**c**) Maximum amplitude projection images of the breast—(**i**) sagittal projection, and (**ii**) coronal projection.

**Figure 9 jcm-11-01165-f009:**
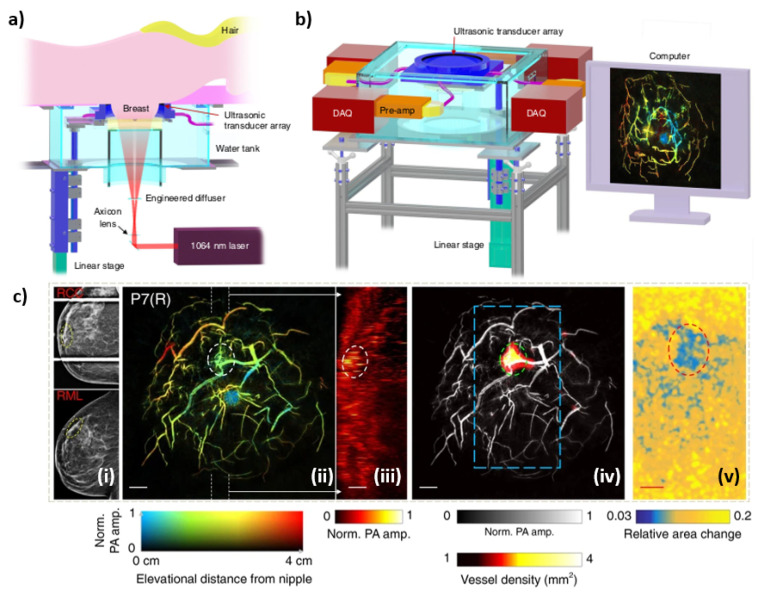
(**a**) Overview of the SBH-PACT system. (**b**) Perspective view of the system with patient bed and optical components removed. DAQ: data acquisition system, Pre-amp: pre-amplifier circuits. (**c**) X-ray and photoacoustic (PA) images of a 44-year-old female patient with a fibroadenoma in the right breast—(**i**) X-ray mammograms of the affected breasts, (**ii**) Depth-encoded angiograms, (**iii**) Maximum amplitude projection images of thick slices in sagittal planes marked by white dashed lines in (**ii**), (**iv**) Automatic tumor detection on vessel density maps, and lastly (**v**) PA elastography images.

**Figure 10 jcm-11-01165-f010:**
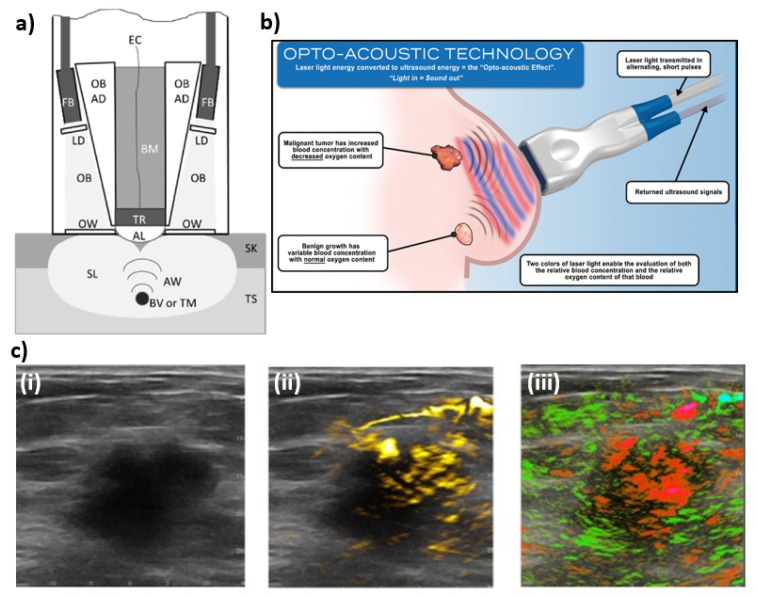
(**a**) Schematic diagram of the Imagio system. Tissue (TS), skin (SK), scattered light (SL), optical beams (OB), fiber bundles (FB), light diffusers (LD), optical windows (OW), acoustic waves (AW), blood vessels or tumors (BV or TM), acoustic lens (AL), transducers (TR), electrical cables (EC), backing material (BM). (**b**) Illustration shows that laser light emitted at wavelengths corresponding to absorption peaks of oxygenated and deoxygenated hemoglobin produces acoustic signals that can then be used to reconstructed oxygen saturation maps. (**c**) An example of combined ultrasound/photoacoustic images of breast carcinoma—(**i**) An ultrasound gray scale image of a 2.6 cm malignant mass, (**ii**) regions of increased total hemoglobin, and (**iii**) oxygenation map where red are regions below an average oxygen saturation of 85% while green are normally oxygenated regions (>90% sO_2_).

**Figure 11 jcm-11-01165-f011:**
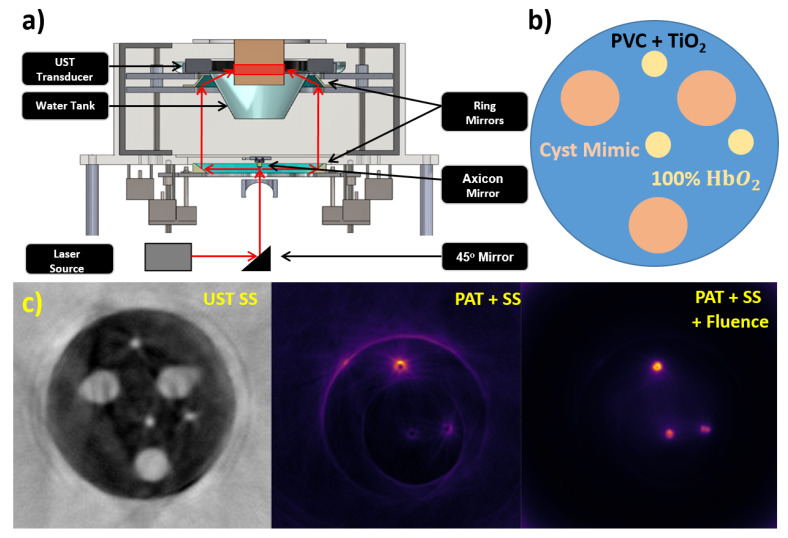
Our USPAT system prototype (**a**) Full system schematic, (**b**) heterogeneous phantom structure, (**c**) original SS image used for compensations, PAT image with SS compensation, PAT image with SS and fluence compensation. PVC: polyvinyl chloride, HbO_2_: oxygenated hemoglobin.

**Table 1 jcm-11-01165-t001:** Overview of differences between transmission ultrasound tomography (TUST) systems.

	Delphinus	QT Ultrasound	Mastoscopia	Karlsruhe
Geometry	Ring	Opposing arrays (one linear, one 2D)	Opposing arrays (2 sets of linear arrays on each side)	Hemisphere
Transmitter number	2048 (sequential)	256 plane wave	N/A	628 (sequential)
Receiver number	2048	2048	N/A	1413
Central frequency	2.5 MHz	0.9 MHz	2 MHz	2.5 MHz
Volumetric imaging?	Through vertical translation	Through vertical translation	Through vertical translation	Inherently 3D
Breast density stratification	Yes	Yes	No	No
Output images	Reflectivity, SS, attenuation	Reflectivity, SS, attenuation	Probability of malignancy	Reflectivity, SS, attenuation
Reflection mode Resolution	0.125 mm	0.96 mm	N/A	0.24 mm
SS Resolution	0.25 mm	1.49 mm	N/A	0.24 mm
FDA status	Approved	Cleared *	N/A	N/A

* United States Food and Drug Administration (FDA) 510 (k) clearance.

**Table 2 jcm-11-01165-t002:** Comparison between major existing breast photoacoustic tomography (PAT) systems.

	PAM-02	Twente PAM 2	OptoSonics	LOUISA-3D	SBH-PACT	Imagio
Transducer/ Detector Geometry	2D Matrix Array	Hemisphere	Spherical	Arc	Ring	Linear Array
Receiver number	600 (20 × 30)	384	512	96	512	128
Central frequency	2 MHz	1 MHz	2 MHz	50 kHz—6 MHz frequency range	2.25 MHz	0.1–12 MHz frequency range
Bandwidth	130%	100%	70%	N/A	95%	N/A
Illumination Geometry	Rectangles Behind and in-front of PA detector	9 fiber bundles	Cone beam	Arc	Donut	Two Rectangular Windows
Laser Wavelengths	700–900 nm	755 and 1064 nm	750–800 nm	757, 797 nm	750, 1064 nm	757, 1064 nm
Resolution	1 mm	1.06 mm (lateral) 0.96 mm (elevational)	0.42 mm	−0.3 mm	0.255 mm	0.73–0.81 mm (lateral) 0.42–0.47 mm (axial)
Volumetric Imaging	Yes, via rotation	Yes	Yes	Yes, via rotation	Yes, via linear scanning	No
Breast Compression	Yes	No	No	No	Yes	No
Co-registered with US	Yes	No	No	Yes	No	Yes
FDA Status	N/A	N/A	N/A	N/A	N/A	Approved

## Data Availability

Not applicable.

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
