# Peer review of "Ultrasound and Photoacoustic Imaging of Breast Cancer: Clinical Systems, Challenges, and Future Outlook"

_jcm, 2022, doi:10.3390/jcm11051165_

Round 1

Reviewer 1 Report

The authors resolved most of my concerns. 

However, I still feel the system, USPAT (BUSPAT), in section 5 should be listed in section 3, that will provide a more objective review. 

In section 4 Line [491-492], iRFP and Evans blue were listed but they didn't be discussed in the following paragraph. 

A comparison table of exogenous contrast agents will be helpful for the readers to follow this section.

Reviewer 2 Report

Thank you for addressing all the comments and making the relevant changes in the manuscript.

Reviewer 3 Report

The manuscript was revised following reviewer comments. I agree to the publication of this manuscript.
